# Self-selection biases in psychological studies: Personality and affective disorders are prevalent among participants

Izabela Kaźmierczak[1]⊗*, Anna Zajenkowska[1]⊗, Radosław Rogoza[2,3], Peter K. Jonason[4], Dawid Ścigała[1]

1 Institute of Psychology, Maria Grzegorzewska University, Warsaw, Poland, 2 Institute of Psychology, Cardinal Stefan Wyszyński University, Warsaw, Poland, 3 Social Innovation Chair, University of Lleida, Lleida, Spain, 4 Department of General Psychology, University of Padova, Padova, Italy

⊗ These authors contributed equally to this work.
* ikazmierczak@aps.edu.pl

**Data Availability Statement:** All data files for this study are available from the Open Science Foundation database (https://osf.io/83usj/).

## Abstract

Respondents select the type of psychological studies that they want to participate in consistence with their needs and individual characteristics, which creates an unintentional self-selection bias. The question remains whether participants attracted by psychological studies may have more psychological dysfunctions related to personality and affective disorders compared to the general population. We investigated (N = 947; 62% women) whether the type of the invitation (to talk about recent critical or regular life events) or the source of the data (either face-to-face or online) attracts people with different psychopathology. Most importantly, participants who alone applied to take part in paid psychological studies had more symptoms of personality disorders than those who had never before applied to take part in psychological studies. The current results strongly translate into a recommendation for either the modification of recruitment strategies or much greater caution when generalizing results for this methodological reason.

## Introduction

Modern psychology research relies on in-person and online methods of data collection where participants can be paid or volunteer their time. Ostensible participants have the right to choose among these options but it is unclear what motivates these choices beyond basic characteristics [1, 2] and a few scant studies on personality traits like extraversion [3], approval-seeking [4], political conservatism [5], time perspective [6], and sensation seeking [7]. Also the type of reward (e.g., lottery reward versus charity reward) attracts participants with different values and further this self-selection results in differences in the task outcomes [2]. In this study we try to provide a better understanding of the role of personality and affective pathologies have on self-selection biases in psychological research.

Generally, self-selection may create unintentional participant × study topic/method biases [5]. Based on this effect, Carnahan and McFarland [8] questioned whether participant's self-

**Funding:** "To conduct Face-to-Face Studies IK was supported by grants 2017/01/X/HS6/02022 from the National Center of Science and BNS 52/20-P. PKJ was partially funded by the grant 2019/35/B/HS6/00682 from National Center of Science. The Online Survey with Volunteers Sample was supported by Ministry of Science and Higher Education in Poland in the form of subsidy for the maintenance and development of research potential at The Maria Grzegorzewska University in 2020. The funders had no role in study design, data collection and analysis, decision to publish, or preparation of the manuscript."

**Competing interests:** The authors have declared that no competing interests exist.

selection could have led to the cruelty in the Stanford Prison Experiment (SPE), which would cast doubt on the generalizability of the well-known findings. The creative experiment started with the advert that invited college students to take part either in "a psychological study of prison life" or just in a psychological study [8] pp. 604). Individuals higher on aggressiveness, authoritarianism, Machiavellianism, narcissism, and social dominance were more attracted by the idea to try "prison life" than those who were more empathic and altruistic.

Volunteer biases and personality traits that motivate people to participate in certain kinds of face-to-face research may affect the collected data. On the other hand, modern psychological scientists are increasingly reliant on online data collection methods. Online surveys are quite appealing because they enable data collection of larger numbers of participants within a short time [9]. Generally speaking, (e.g., considering internal motivation, enjoyment-seeking and extraverted behaviors), online samples appear sufficiently similar to offline samples [10–12]. However, there is also some evidence of greater depression and social isolation levels among online participants [13], but this effect may have waned as more and more of the people's lives move online.

Conceivably some individuals willing to take part in psychological studies are seeking a therapeutic environment, diagnoses, and/or a meeting with a psychologist alike with participants who declare higher self-reported external eating tendencies and positive preoccupations with food choosing an eating-related study [6]. Therefore, participants of psychological studies may have more psychological dysfunctions compared to the general population. Nowadays psychological problems are experienced by a large group of people with overall rates of personality disorders ranging from 4.4% [14] to 14.8% [15]. The wide range and the differences between studies mostly derive from methodology [16], however they might also be related to the characteristics of participants who decide to take part in a survey. Still, visiting a psychiatrist or psychologist is a burden to many people and participating in a psychological study might be perceived as a cheap substitute or alternative to acquire some professional help.

The current project focuses on self-selection bias related to personality disorders in different types of studies. We analyzed personality disorders according to both dimensional (i.e., number of symptoms) and categorical (i.e., number of diagnosed personality disorders) models, and additionally we applied the measure of borderline personality organization. The latter combines both aforementioned models of borderline personality disorder (BPD), and the dimensional approach to BPD can also indicate the intensity of the general pathology of personality [17].

Consequently, the project has two main aims and investigates (1) whether the type of the invitation (to talk about recent critical or regular life events) or (2) the source of the data (either face-to-face or online) attract participants with different or profounder personality disorders. We also (3) compared the groups to individuals who had never participated in any psychological studies (non-volunteers). Moreover, because of the comorbidity of personality disorders with depression and anxiety symptoms and the specificity of online samples we also tested (4) whether participants who enroll for online psychological studies are higher in depressive and anxiety symptoms than those who had never done so.

We hypothesized that all volunteers, and particularly the participants of the studies on critical life event, which is normatively less expected life experience [18] and leads to comorbid disorders [19], would present more pathological personality organization (i.e., number of symptoms as well as diagnosed personality disorders) as compared to non-volunteers (*Hypothesis 1*). We also predicted that online panel volunteers would show greater pathological personality organization and more emotional disturbances than non-volunteers (*Hypothesis 2*).

## Materials and methods

### Participants & procedure

**Face-to-face studies with volunteers sample and experimental manipulation.** Participants were volunteers ($N$ = 127; 58.3% women) between 21 and 41 years of age ($M$ = 27.79, $SD$ = 5.58), who responded to an online advertisement by writing an email to schedule an individual meeting. The advertisements were disseminated on the same websites with paid survey offers; however, we advertised the study using three different forms of the invitation (please see S1 File) as an experimental manipulation. Participants could respond either to an advertisement in relation to a negative critical life event (CLE, such as parting with a partner, serious illness, death of a close family member, or losing a job) and its psychological consequences that took place up to two months before the research and (a) led to low mood (CLElm Group, $n$ = 32; The depressed mood criterion of the advertised study was a part of an experimental manipulation. Participants themselves decided if they were depressed in the aftermath of a critical life event or not.), or (b) did not necessarily lead to low mood (CLE Group, $n$ = 45; In this experimental condition there was no information in the advertisement on depressed mood as a criteria to participate in the study.), or (c) to an advertisement regarding a study about regular (typical) life events (RLE) and their psychological impact (RLE Group, $n$ = 50). Nearly half (50.7%) of the participants had a university degree, 28% were undergraduate/postgraduate students, and 18.7% participants completed high school. Respondents were paid for their participation with vouchers (€35–50). No studies in this manuscript were preregistered. We report all manipulations, measures, and exclusions in this article.

**Face-to-face study with non-volunteers sample.** The non-volunteer sample were participants ($N$ = 100; 56% women) between 20 and 48 years of age ($M$ = 29.67, $SD$ = 6.17) who had never enrolled to take part in any paid psychological studies before. They were approached by experimenters within their local communities. Most of them (66%) had a Bachelor's or Master's degree, 23% were undergraduate/postgraduate students, and 11% participants completed high school.

**Online survey with volunteers sample.** Participants were volunteers ($N$ = 720; 71.9% women) between 25 and 45 years of age ($M$ = 34.37, $SD$ = 5.71). The study was carried out on the Internet by a nationwide research panel located in Poland. Each volunteer collects points for each study he/she participates in within a timeframe. The points are summed up and can be exchanged for a prize (e.g., a book, headphones, household appliances) chosen from the list. 56.7% of online panel participants had a Bachelor's or Master's degree, 1.9% were undergraduate/postgraduate students, 33.2% participants completed high school, 5.7% of them had vocational education, and 2.3% had either primary or lower secondary education.

In all studies, participants were from Poland and self-reported they were ethnically "white". None of the observations were excluded. The samples size allows for detection of an effect size $f$ = .25, $\alpha$ = .05 with a power of .80 according to the power analysis in G*Power 3.1 software [20].

After recruitment, which was specific and group dependent as described above, participants followed the same procedure. They were informed of the nature of the study they were partaking and provided their consent. Regardless of the group, they completed the same series of self-report measures. Only the measurement of individual differences in personality disorders was restricted to face-to-face studies because its procedure requires the presence of researcher. Upon completion they were thanked, debriefed, and rewarded where relevant. Face-to-face studies were conducted in 2019 and 2020 before the COVID-19 pandemic, respectively, while the online survey took place in 2020 during the COVID-19 pandemic.

The project was approved by the Academic Human Research Ethics Committee at the Maria Grzegorzewska University (159-2017/2018 and 183-2018-2019).

## Measures

**Structured clinical interview SCID-II for DSM-IV [21].** To assess individual differences in personality disorders we used the Polish version of the measure [22]. Specifically, the psychological diagnoses were made using the Screening Modules for Axis II Disorders. The tool is a self-report questionnaire based on the DSM criteria for each of 12 personality disorders (PDs; e.g., Narcissistic, Avoidant or Antisocial Personality Disorder). Participants are asked whether (yes/no) 119 items (e.g., Do you often worry that in social situations someone will criticize or reject you?) apply to them. Each disorder is scored continuously based on a count of the affirmative responses which are then dichotomized based on DSM cutoff protocols [22]. Both continuous and dichotomized indices were used in our studies. The tool has satisfactory reliability (e.g., for all PDs mean *kappa* = .80 in [23] and validity [24, 25].

**Borderline personality inventory [26].** To measure individual differences in borderline personality organization, we used the Polish translation of the measure [27] The short version consists of 20 dichotomous items (true or false). It is based on Kernberg's [28] concept of borderline personality, but the diagnostic criteria are compatible with both the DSM-IV and Gunderson and Kolb's [29] concept of BPD. In our studies, items were summed (Cronbach's $\alpha$ = .87) and had good internal consistency as with previous research [30].

**Hospital anxiety and depression scale [31].** To capture individual differences in depression and anxiety (within the last week) we used the Polish translation of the measure [32]. The questionnaire consists of 14 items (7 per each trait) where participants were asked to read each of them and mark the appropriate answer that came closest to how they had felt during the last week. In both cases, items were summed and consistent with previous research [33], the internal consistency for the depression ($\alpha$ = .86) and anxiety ($\alpha$ = .80) aspects were good.

## Results

To test *Hypothesis 1* referring to the differences in the level of personality disorders across the groups of volunteers and a group of non-volunteers, a one-way ANOVA with Bonferroni *post hoc* tests (with alpha level set to .05) was conducted. The results are given in Table 1.

The analyzed groups differed in the number of symptoms of all personality disorders except for Histrionic. Results revealed that asking for low mood in the invitation (i.e., CLElm Group) "attracts" most pathological participants (i.e., number of symptoms as well as diagnosed personality disorders), while Non-Volunteers had the least symptoms than any other group. No mood criterion in research advertisements resulted in more symptoms than the RLE and Non-Volunteer groups, but there were no differences in the overall diagnosed personality disorders.

Participants from CLElm Group had more symptoms of disorders of Avoidant, Dependent, Passive-Aggressive, Schizoidal, and Narcissistic PDs than from the RLE and Non-Volunteers Groups, but not from those who were asked for critical event without information on mood. The CLElm Group also differed in the levels of Schizotypal, Paranoidal, Antisocial, and Obsessive-Compulsive PDs, however only in relation to Non-Volunteers. Participants from CLE Group scored higher on Schizoidal PD than RLE and Non-Volunteers Groups. Non-Volunteers scored lower than any other group on Obsessive-Compulsive and Borderline PDs.

Additionally, to test the differences in meeting personality disorders criteria across surveyed groups, $\chi^2$ tests were conducted (Table 2).

We found differences in the number of diagnoses in Avoidant, Dependent, Obsessive-Compulsive, Passive-Aggressive, Narcissistic, and Borderline PDs. In all these groups, most diagnoses were proportionally assigned to the CLElm Group.

To test *Hypothesis 2* referring to the differences between Non-Volunteers and online panel participants, two *t*-tests for independent samples were conducted. Non-Volunteers scored

**Table 1. Results of the one-way ANOVA across different groups of volunteers and non-volunteers.**

| Personality disorder | Group | *M* | *SD* | *F* | partial η² |
|---|---|---|---|---|---|
| Avoidant | CLE | 2.34 | 2.19 | 6.72** | .09 |
| | CLElm | 3.32 | 2.25 | | |
| | RLE | 1.62 | 1.56 | | |
| | Non-Volunteers | 1.59 | 1.95 | | |
| Dependant | CLE | 2.20 | 1.58 | 5.98** | .08 |
| | CLElm | 3.21 | 1.71 | | |
| | RLE | 1.82 | 1.52 | | |
| | Non-Volunteers | 1.84 | 1.61 | | |
| Obsessive-Compulsive | CLE | 4.15 | 1.94 | 10.17** | .12 |
| | CLElm | 4.89 | 1.91 | | |
| | RLE | 4.04 | 1.80 | | |
| | Non-Volunteers | 3.68 | 1.97 | | |
| Passive-Aggressive | CLE | 3.00 | 2.04 | 7.54** | .10 |
| | CLElm | 3.89 | 1.85 | | |
| | RLE | 1.90 | 1.69 | | |
| | Non-Volunteers | 2.27 | 2.09 | | |
| Paranoid | CLE | 2.68 | 2.20 | 3.23* | .04 |
| | CLElm | 3.68 | 2.39 | | |
| | RLE | 2.54 | 1.97 | | |
| | Non-Volunteers | 2.32 | 1.93 | | |
| Schizotypal | CLE | 2.44 | 1.88 | 3.71* | .05 |
| | CLElm | 3.18 | 2.57 | | |
| | RLE | 2.26 | 2.19 | | |
| | Non-Volunteers | 1.79 | 1.84 | | |
| Schizoidal | CLE | 1.90 | 1.30 | 8.17** | .10 |
| | CLElm | 2.25 | 1.43 | | |
| | RLE | 1.20 | 1.14 | | |
| | Non-Volunteers | 1.17 | 1.19 | | |
| Histrionic | CLE | 1.98 | 1.62 | 0.19 | .00 |
| | CLElm | 1.79 | 1.73 | | |
| | RLE | 2.08 | 1.74 | | |
| | Non-Volunteers | 1.93 | 1.76 | | |
| Narcissistic | CLE | 4.17 | 3.17 | 5.39** | .07 |
| | CLElm | 5.93 | 3.14 | | |
| | RLE | 3.90 | 2.67 | | |
| | Non-Volunteers | 3.36 | 3.07 | | |
| Borderline | CLE | 4.83 | 3.61 | 13.81** | .16 |
| | CLElm | 7.32 | 3.66 | | |
| | RLE | 4.48 | 3.27 | | |
| | Non-Volunteers | 2.90 | 3.16 | | |
| Antisocial | CLE | 1.32 | 1.59 | 3.91* | .06 |
| | CLElm | 1.59 | 1.53 | | |
| | RLE | 0.80 | 1.08 | | |
| | Non-Volunteers | 0.80 | 1.14 | | |
| Number of Symptoms | CLE | 31.00 | 15.46 | 18.89** | .22 |
| | CLElm | 39.81 | 12.69 | | |
| | RLE | 26.47 | 12.30 | | |

*(Continued)*

**Table 1.** (Continued)

| Personality disorder | Group | *M* | *SD* | *F* | partial η² |
|---|---|---|---|---|---|
| | Non-Volunteers | 20.25 | 11.30 | | |
| Number of PDs | CLE | 3.45 | 3.01 | 12.13** | .15 |
| | CLElm | 5.71 | 3.04 | | |
| | RLE | 2.62 | 2.17 | | |
| | Non-Volunteers | 3.14 | 2.75 | | |

*Note.* We also assessed a two-factor ANOVA to assess whether there were gender differences, however we did not find group × gender interactions and moreover, the calculated estimates were largely underpowered.

* $p < .05$

** $p < .01$

lower ($t_1[817] = -7.14$, $p_1 < .001$, Cohen's $d = -0.22$; $t_2[818] = -7.43$, $p_2 < .001$, $d = -0.25$) on anxiety ($M_1 = 11.06$, $SD_1 = 4.28$) and depression ($M_2 = 10.07$, $SD_2 = 4.20$) symptoms than did the online panel participants ($M_1 = 14.08$, $SD_1 = 3.89$; $M_2 = 13.35$, $SD_2 = 4.13$).

Additionally, one-way ANOVAs were conducted on all groups (i.e., CLE, CLElm, RLE, Non-Volunteers, and online panel participants) simultaneously, to assess the differences in the organization of borderline personality (using BPI, which at the same time allows for both dimensional and categorical approach to BPD; Leichsenring, 1999). We did not find any differences across studied groups ($F[4, 929] = 1.98$, $p = .096$; partial $\eta^2 = .01$) in the intensity of symptoms, however, we found differences in the number of diagnoses (i.e., score > 10), where CLElm Group most frequently was diagnosed with borderline personality organization ($\chi^2[4] = 12.30$, $p = .015$).

## Discussion

The main aim of this project was to investigate self-selection biases related to the prevalence of personality disorders in psychological studies. We tested whether different types of research invitations attract different research participants in terms of their psychopathology. Indeed, people who replied to an advertisement on a study on a negative critical life event and its psychological consequences that took place up to two months before the research and led them to low mood had not only more personality disorders (PDs), but also the number of symptoms for different types of PDs compared to those who volunteered for a study on a regular life event and non-volunteers. Also, participants who replied to an advertisement on a study on a recent negative critical life event without the low mood requirement had more symptoms than those who volunteered for a study on a regular life event and non-volunteers. Still those who never participated in research before (i.e. non-volunteers) were likely to show the least symptoms of PDs compared to those who did, suggesting that people with the healthiest structure of personality (and reflecting the general population) are not usually included in research samples or are relatively rarely.

At the same time, personality disorders (more numerous and higher in volunteers) are associated with rigid (and maladaptive) beliefs and the resulting inflexible behavioral patterns [34, 35]), which may be of great importance in experimental research, particularly while interpreting the effectiveness of an experimental manipulation, but also in the identification and/or description of any psychological phenomena. Many studies show that participants with PDs demonstrate specific attentional coping styles [36] and biased attention to emotions and facial expressions [37, 38], which might interplay with all experimental procedures.

**Table 2. The number of personality disorders diagnoses across groups.**

| Personality disorder | Group | Criteria not met (%) | % | Criteria met (%) | % | $\chi^2_{(3)}$ |
|---|---|---|---|---|---|---|
| Avoidant | CLE | 26 | 66.7 | 13 | 33.3 | 13.58** |
|  | CLElm | 14 | 50 | 14 | 50 |  |
|  | RLE | 43 | 86 | 7 | 14 |  |
|  | Non-Volunteers | 77 | 77 | 23 | 23 |  |
| Dependant | CLE | 34 | 75.6 | 6 | 13.3 | 9.42* |
|  | CLElm | 20 | 71.4 | 8 | 28.6 |  |
|  | RLE | 47 | 94 | 3 | 6 |  |
|  | Non-Volunteers | 90 | 90 | 10 | 10 |  |
| Obsessive-Compulsive | CLE | 15 | 37.5 | 25 | 62.5 | 17.00** |
|  | CLElm | 7 | 25 | 21 | 75 |  |
|  | RLE | 32 | 64 | 18 | 36 |  |
|  | Non-Volunteers | 60 | 60 | 40 | 40 |  |
| Passive-Aggressive | CLE | 26 | 65 | 14 | 35 | 11.77** |
|  | CLElm | 12 | 42.9 | 16 | 57.1 |  |
|  | RLE | 40 | 80 | 10 | 20 |  |
|  | Non-Volunteers | 70 | 70 | 30 | 30 |  |
| Paranoid | CLE | 25 | 62.5 | 15 | 37.5 | 4.47 |
|  | CLElm | 15 | 53.6 | 13 | 46.4 |  |
|  | RLE | 37 | 74 | 13 | 26 |  |
|  | Non-Volunteers | 71 | 71 | 29 | 29 |  |
| Schizotypal | CLE | 34 | 85 | 6 | 15 | 6.63 |
|  | CLElm | 19 | 67.9 | 9 | 32.1 |  |
|  | RLE | 42 | 84 | 8 | 16 |  |
|  | Non-Volunteers | 88 | 88 | 12 | 12 |  |
| Schizoidal | CLE | 36 | 90 | 4 | 10 | 7.67 |
|  | CLElm | 22 | 78.6 | 6 | 21.4 |  |
|  | RLE | 43 | 86 | 7 | 14 |  |
|  | Non-Volunteers | 95 | 95 | 5 | 5 |  |
| Histrionic | CLE | 37 | 92.5 | 3 | 7.5 | 0.26 |
|  | CLElm | 25 | 89.3 | 3 | 10.7 |  |
|  | RLE | 46 | 92 | 4 | 8 |  |
|  | Non-Volunteers | 91 | 91 | 9 | 9 |  |
| Narcissistic | CLE | 25 | 62.5 | 15 | 37.5 | 15.89** |
|  | CLElm | 8 | 28.6 | 20 | 71.4 |  |
|  | RLE | 31 | 62 | 19 | 38 |  |
|  | Non-Volunteers | 70 | 70 | 30 | 30 |  |
| Borderline | CLE | 22 | 55 | 18 | 45 | 27.00** |
|  | CLElm | 8 | 28.6 | 20 | 71.4 |  |
|  | RLE | 24 | 48 | 26 | 52 |  |
|  | Non-Volunteers | 77 | 77 | 23 | 23 |  |
| Antisocial | CLE | 30 | 75 | 10 | 25 | 3.74 |
|  | CLElm | 17 | 60.7 | 11 | 39.3 |  |
|  | RLE | 40 | 80 | 10 | 20 |  |
|  | Non-Volunteers | 70 | 70 | 30 | 30 |  |

* $p < .05$

** $p < .01$

Comparing the studied groups to the general population, it should be noted that both the ratio of participants who met the criteria of a PD (from six to 75% depending on a type of PDs and an advertised study) and the number of clinically diagnosed PDs (from three to six coexisting PDs dependent on a type of an advertised study) are unexpected outcomes. All comparison groups differed to a greater or lesser extent from the distribution of PDs in the general population, regardless the wide range of the results. Studies show that it ranges from 4.4% to 13.4% [14, 39] for the European population and from 9.0% to 21.5% [3, 40] for the United States population. The highest overall prevalence of PDs (equal to 45.5%) has been identified amongst psychiatric patients [31], and it is still lower than in one of our advertised study (i.e., after a critical life event that took place up to two months before the research and led to low mood).

In addition, the prediction that the personality organization of both online participants and volunteers who applied for different types of face-to-face studies will be more pathological in comparison with non-volunteers was verified in the project. However, there were no differences in the intensity of Borderline PD symptoms, nonetheless volunteers who participated in a critical life event study and with low mood were diagnosed with this disorder most frequently. Although such key words included in the research invitation as "low mood" and "negative critical life event" had the power to attract people with particularly increased personality psychopathologies, it should be noted that all volunteers, regardless the experimental group, were characterized with its higher level. It might suggest that apart from participation in a study, they might indirectly seek for a psychological help and for a reason, however this hypothesis requires further investigation.

Furthermore, online participants were higher on depression and anxiety (their mean scores indicate clinical "caseness" using the cut-off of 11 points suggested by Zigmond and Snaith [31] as compared to those who never participated in research before (their mean scores indicate a borderline level for depression and clinical "caseness" for anxiety; [31] This finding is in line with some previous studies [13], however it is important to acknowledge that the online survey was conducted during COVID-19, which might have an aversive impact on participants. As Bueno-Notivol et al. [41] showed in their meta-analysis, the prevalence of depression (25%) in COVID-19 is even seven times higher compared to its global estimation in 2017. At the same time, although it causes a methodological restriction to adequately compare all groups, we cannot ignore the fact that all data collected during the pandemic is based on this specific research samples (e.g., most research is done remotely). Hence, paradoxically, the Internet sample from our study delivers a characteristic of the subjects who we are actually being studied now.

Additionally, our study was conducted in the early stages of the COVID-19 pandemic. This means that the heightened state of anxiety that emerged in almost everyone was not able to alter enduring dispositional personality traits. As longitudinal studies have shown, change is a dynamic and temporal process. A longitudinal study of Caldioroli et al. [42] involving 166 individuals affected by different psychiatric disorders at three time points ($t_0$ as pandemic outbreak, $t_1$ as lockout period, $t_2$ as re-opening) showed significant deterioration during the lockout period with little improvement during the re-opening. Moreover, only psychopathology in patients with schizophrenia and obsessive-compulsive symptoms were not significantly improved at $t_2$. Individuals with PDs were at higher risk for overall psychopathology than those with depression and anxiety/obsessive-compulsive and exhibited more severe anxiety symptoms than schizophrenic patients.

Summing up, as (1) volunteers vary in terms of psychopathology depending on the type of both invitation and study they wanted to participate in, and also differ from those individuals who usually do not come to psychological research, (2) there is no basis for assuming that the

presented findings are an isolated case. Hence, it is advisable to interpret all psychological research outcomes considering the impact of the form of invitation to the research and the type of research itself on the potential psychopathology of the participants. Moreover, the research outcomes need to be interpreted in close connection with the finding of larger psychopathology of the volunteers compared to non-volunteers. This conclusion translates into a recommendation for either the modification of recruitment strategies or much greater caution when generalizing results for this methodological reason.

## Conclusions

Despite the unique nature of these studies, there are several limitations worth considering. First, all the samples were of Poles which may undermine the generalizability of the findings. Subsequent work should adopt cross-national samples to test for potential moderation effects. Second, the effects might still be localized to particular areas of psychology given that we needed to adopt particular methods and participants had at least nominal information about the methods that would be used a la the informed consents. Future research may need to actively deceive those solicited to conduct a study to better test our assertions. Third, our results are pinned to the measures we used and thus, future research might adopt a wider range of traits like including the Dark Tetrad of psychopathy, narcissism, sadism, and Machiavellianism in their subclinical forms or the Personality Inventory for DSM-5. Another important dimension to consider is the participants' level of self-esteem, as this could be an important factor in determining the willingness to participate in the study.

We have concerned ourselves with a fundamental concern in research in several areas of psychology. Researchers often take for granted that the way they advertise their studies and who they recruit do not appreciably affect their outcomes. In our studies, we have shown that those who have more personality pathologies are more drawn to studies where they can express their trauma and may be simply more likely to volunteer for studies. While we cannot dismiss the fact that all our samples were of Poles, if we assume that they are like others around the world—an assumption we see no reason to doubt, which is, after all, the null hypothesis—then our results have meaningful implications for how researchers interpret their results and how clinicians estimate the prevalence of various disorders. In short, our field may be conducting research on an atypically disordered and motivated group of people leading to biased views of the reality of psychological effects.

Now that we have revealed some serious implications for the conclusions we draw from typical research participants, the next logical question is what can be done about it. We propose three solutions that should not be too onerous. First, we suggest alternative recruitment strategies. For instance, researchers could expressly seek participants who have not participated in research before or include, as a demographic question, how many times in the last month (for instance) a participant took a survey. This might even be something sites like Prolific or Ariadna could provide as part of their meta-data about participants. Second, researchers can engage samples that are more general than the typical psychology student samples including the various Facebook groups for gathering quick data. Third, if the concern is that those who take surveys too often might have psychopathologies, controlling for them with some measure of broadband psychopathologies like the DSM-5 Brief Form PID-5 ((PID-5-BF, American Psychiatric Association, 2013) might be useful. It would allow researchers to ensure that effects found—whether they be correlational or experimental—were not spuriously driven by some psychopathologies. Collectively, these practices would increase the veracity and trustworthiness of findings in much of psychology. While we understand these steps might be

annoying and potentially bloating to one's methods a bit, we think the trade-off between in time/effort is worth what will be gained in accuracy.

## Supporting information

**S1 File. Methodology file.**
(DOCX)

## Acknowledgments

**Code availability** (software application or custom code): IBM SPSS Statistics 25.

**Informed Consent**: Interested volunteers were informed about the nature and purpose of the study and offered the opportunity to participate. When they chose to participate, they were informed that they could discontinue at any time and their responses would be confidential and not revealed to anyone. In Face-to-Face Studies participants provided their written informed consent to participate. In Online Survey with Volunteers Sample only people who chose to register for the research pool were able to take part (an invitation was sent to them by the pool mailing system). Interested volunteers were informed about the nature and purpose of the study and offered the opportunity to participate. When they chose to participate, they were informed that they could discontinue at any time and their responses would be confidential and not revealed to anyone. They provided informed consent by clicking "Participate" having read the invitation on the research platform.

**Preregistration:** Current study was not preregistered.

## Author Contributions

**Conceptualization:** Izabela Kaźmierczak, Anna Zajenkowska.

**Data curation:** Izabela Kaźmierczak.

**Formal analysis:** Radosław Rogoza, Dawid Ścigała.

**Funding acquisition:** Izabela Kaźmierczak, Anna Zajenkowska, Peter K. Jonason.

**Methodology:** Izabela Kaźmierczak, Anna Zajenkowska.

**Project administration:** Izabela Kaźmierczak.

**Supervision:** Peter K. Jonason.

**Writing – original draft:** Izabela Kaźmierczak, Anna Zajenkowska, Dawid Ścigała.

**Writing – review & editing:** Izabela Kaźmierczak, Anna Zajenkowska.

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
