## [Decision Letter · Decision Letter 0]

11 Nov 2022

PONE-D-22-27998Self-selection biases in psychological studies Personality and affective disorders are prevalent among participantsPLOS ONE

Dear Dr. Kaźmierczak,

Thank you for submitting your manuscript to PLOS ONE. After careful consideration, we feel that it has merit but does not fully meet PLOS ONE’s publication criteria as it currently stands. Therefore, we invite you to submit a revised version of the manuscript that addresses the points raised during the review process.

We look forward to receiving your revised manuscript.

Kind regards,

Ankit Jain, M.D.

Academic Editor

PLOS ONE

Journal Requirements:

a) Did participants provide their written or verbal informed consent to participate in this study?

"Funding: The first author was supported by grants 2017/01/X/HS6/02022 from the National

Center of Science and BNS 52/20-P. The second author was supported by the grant UMO2017/26/D/HS6/00258 from the National Center of Science and the fourth author was

partially funded by the grant 2019/35/B/HS6/00682 from National Center of Science. The

online study was supported by Ministry of Science and Higher Education in Poland in the

form of subsidy for the maintenance and development of research potential at The Maria

Grzegorzewska University in 2020."

"IK was supported by grants 2017/01/X/HS6/02022 from the National Center of Science and BNS 52/20-P. 

AZ was supported by the grant UMO-2017/26/D/HS6/00258 from the National Center of Science.

PKJ was partially funded by the grant 2019/35/B/HS6/00682 from National Center of Science. 

The online study was supported by Ministry of Science and Higher Education in Poland in the form of subsidy for the maintenance and development of research potential at The Maria Grzegorzewska University in 2020.

Reviewers' comments:

Reviewer's Responses to Questions

**Comments to the Author**

1. Is the manuscript technically sound, and do the data support the conclusions?

Reviewer #1: Yes

Reviewer #2: Yes

Reviewer #3: Yes

2. Has the statistical analysis been performed appropriately and rigorously? 

Reviewer #1: N/A

Reviewer #2: Yes

Reviewer #3: Yes

3. Have the authors made all data underlying the findings in their manuscript fully available?

Reviewer #1: Yes

Reviewer #2: Yes

Reviewer #3: Yes

4. Is the manuscript presented in an intelligible fashion and written in standard English?

Reviewer #1: Yes

Reviewer #2: Yes

Reviewer #3: Yes

5. Review Comments to the Author

Reviewer #1: This is an interesting manuscript to review as it highlights that there are significant ways in which research studies are distinct from one another, and these differences have the potential to affect the allure of various studies to various individuals.

The authors try to point out that it is unclear as to the potential personality and affective disorders that may be associated with such choices because people choose the types of psychological studies in which they are willing to participate based on their needs and individual characteristics. This creates an unintentional self-selection bias, but it is important to note that this bias exists regardless.

Most importantly, people who took part in sponsored studies exhibited a greater number of symptoms associated with personality disorders than those who had never before participated for any kind of research.This is a very interesting finding and should prompt researchers to either the alteration of recruiting tactics or much greater caution when generalizing results for this methodological reason.

Reviewer #2: This study presents an interesting concept with a good research question and interesting implications for current research being done in the field of psychiatry. Comorbid personality disorder symptoms can make mood disorders difficult to treat, with patients less likely to respond to medications, more likely to face socioeconomic challenges, likely to suffer from mood symptoms for longer periods of time and hence less likely to comply with their medications. With the most recent research regarding newer antidepressants showing a decline in response to medications and increase in placebo response, this study provides another hypothesis that the patients who are volunteering for these studies may be suffering from additional personality disorder symptoms, which often makes it harder to treat depression.

Overall, I found that the study utilized a good method, and the study design appears clear and easy to replicate. The data analysis appears well done, and there does not appear to be any bias or error.

The authors can potentially identify actionable points in the conclusion section that institutions and other health care providers can utilize in their own research in order to address challenges posed by these findings.

Reviewer #3: It was an interesting concept and would be helpful to know more about such kind of biases that might affect volunteers as a whole. It was helpful to have your excellent statistical analysis. The timing especially with comparing non-Covid times to Covid times especially for a volunteer psychology projects seems like it would have added additional confounding factors to the outcome as you have pointed out. It would be interesting to repeat such a study at the same time and with fewer variables being measured.

6. PLOS authors have the option to publish the peer review history of their article (what does this mean?). If published, this will include your full peer review and any attached files.

Reviewer #1: No

Reviewer #2: **Yes: **Lakshit Jain MD

Reviewer #3: No

---

## [Author Response · Author response to Decision Letter 0]

10 Dec 2022

Dear Ankit Jain, M.D.

Academic Editor of PLOS ONE 

We would like to thank you for the feedback and helpful suggestions from you and your reviewers. We have complied with all of them. We hope that our manuscript is substantially improved and that you and the reviewers now will find it acceptable for publication. All changes in the manuscript are marked in grey.

Below we present replies to all the comments and suggestions.

Self-selection biases in psychological studies: Personality and affective disorders are prevalent among participants

Thank you for style templates. Please see that the manuscript meets PLOS ONE's style requirements at this moment.

a) Did participants provide their written or verbal informed consent to participate in this study? b) If consent was verbal, please explain i) why written consent was not obtained, ii) how you documented participant consent, and iii) whether the ethics committees/IRB approved this consent procedure.

In Face-to-Face Studies participants provided their written informed consent to participate. In Online Survey with Volunteers Sample only people who chose to register for the research pool were able to take part (an invitation was sent to them by the pool mailing system). Interested volunteers were informed about the nature and purpose of the study and offered the opportunity to participate. When they chose to participate, they were informed that they could discontinue at any time and their responses would be confidential and not revealed to anyone. They provided informed consent by clicking “Participate” having read the invitation on the research platform.

"Funding: The first author was supported by grants 2017/01/X/HS6/02022 from the National

Center of Science and BNS 52/20-P. The second author was supported by the grant UMO2017/26/D/HS6/00258 from the National Center of Science and the fourth author was

partially funded by the grant 2019/35/B/HS6/00682 from National Center of Science. The

online study was supported by Ministry of Science and Higher Education in Poland in the

form of subsidy for the maintenance and development of research potential at The Maria

Grzegorzewska University in 2020."

"IK was supported by grants 2017/01/X/HS6/02022 from the National Center of Science and BNS 52/20-P. AZ was supported by the grant UMO-2017/26/D/HS6/00258 from the National Center of Science. PKJ was partially funded by the grant 2019/35/B/HS6/00682 from National Center of Science. The online study was supported by Ministry of Science and Higher Education in Poland in the form of subsidy for the maintenance and development of research potential at The Maria Grzegorzewska University in 2020.The funders had no role in study design, data collection and analysis, decision to publish, or preparation of the manuscript."

 Our amended statement is: "To conduct Face-to-Face Studies IK was supported by grants 2017/01/X/HS6/02022 from the National Center of Science and BNS 52/20-P. PKJ was partially funded by the grant 2019/35/B/HS6/00682 from National Center of Science. The Online Survey with Volunteers Sample was supported by Ministry of Science and Higher Education in Poland in the form of subsidy for the maintenance and development of research potential at The Maria Grzegorzewska University in 2020. The funders had no role in study design, data collection and analysis, decision to publish, or preparation of the manuscript."

 It is done. Both versions are identical now. Thank you.

It is done. Thank you.

The following positions have been deleted from the References section.

1. Beck A, Ward C, Mendelson M, Mock J, Erbaugh J. An inventory for measuring depression. Archives of General Psychiatry. 1961; 4: 561–571.

2. Caldiroli A, Capuzzi E, Tringali A, Tagliabue I, Turco M, et al. The psychopathological impact of the SARS-CoV-2 epidemic on subjects suffering from different mental disorders: An observational retrospective study. Psychiatry Research. 2022; 307: 114334. 

3. Kiejna A, Piotrowski P, Adamowski T, Moskalewicz J, Wciórka J, et al. Rozpowszechnienie wybranych zaburzeń psychicznych w populacji dorosłych Polaków z odniesieniem do płci i struktury wieku–badanie EZOP Polska (Prevalence of selected mental disorders in the population of adult Poles with reference to gender and age structure - EZOP Poland study). Psychiatria Polska. 2015; 49: 15-27.

4. Lyon D, Greenberg J. Evidence of codependency in women with an alcoholic parent: Helping out Mr. Wrong. Journal of Personality and Social Psychology. 1991; 61: 435-439.

5. Wrosch C, Freund AM. Self-Regulation of Normative and Non-Normative Developmental Challenges. Human Development. 2001; 44: 264-283. 

Reviewer #1:

This is an interesting manuscript to review as it highlights that there are significant ways in which research studies are distinct from one another, and these differences have the potential to affect the allure of various studies to various individuals.

The authors try to point out that it is unclear as to the potential personality and affective disorders that may be associated with such choices because people choose the types of psychological studies in which they are willing to participate based on their needs and individual characteristics. This creates an unintentional self-selection bias, but it is important to note that this bias exists regardless.

Most importantly, people who took part in sponsored studies exhibited a greater number of symptoms associated with personality disorders than those who had never before participated for any kind of research. This is a very interesting finding and should prompt researchers to either the alteration of recruiting tactics or much greater caution when generalizing results for this methodological reason.

 Thank you for your feedback.

Reviewer #2

This study presents an interesting concept with a good research question and interesting implications for current research being done in the field of psychiatry. Comorbid personality disorder symptoms can make mood disorders difficult to treat, with patients less likely to respond to medications, more likely to face socioeconomic challenges, likely to suffer from mood symptoms for longer periods of time and hence less likely to comply with their medications. With the most recent research regarding newer antidepressants showing a decline in response to medications and increase in placebo response, this study provides another hypothesis that the patients who are volunteering for these studies may be suffering from additional personality disorder symptoms, which often makes it harder to treat depression. Overall, I found that the study utilized a good method, and the study design appears clear and easy to replicate. The data analysis appears well done, and there does not appear to be any bias or error. The authors can potentially identify actionable points in the conclusion section that institutions and other health care providers can utilize in their own research in order to address challenges posed by these findings.

Dear Dr Jain Lakshit, thank you for your feedback and sharing your medical perspective.

We have added the following paragraph: “Now that we have revealed some serious implications for the conclusions we draw from typical research participants, the next logical question is what can be done about it. We propose three solutions that should not be too onerous. First, we suggest alternative recruitment strategies. For instance, researchers could expressly seek participants who have not participated in research before or include, as a demographic question, how many times in the last month (for instance) a participant took a survey. This might even be something sites like Prolific or Ariadna could provide as part of their meta-data about participants. Second, researchers can engage samples that are more general than the typical psychology student samples including the various Facebook groups for gathering quick data. Third, if the concern is that those who take surveys too often might have psychopathologies, controlling for them with some measure of broadband psychopathologies like the DSM-5 Brief Form PID-5 ((PID-5-BF, American Psychiatric Association, 2013) might be useful. It would allow researchers to ensure that effects found--whether they be correlational or experimental--were not spuriously driven by some psychopathologies. Collectively, these practices would increase the veracity and trustworthiness of findings in much of psychology. While we understand these steps might be annoying and potentially bloating to one's methods a bit, we think the trade-off between in time/effort is worth what will be gained in accuracy”.

Reviewer #3

It was an interesting concept and would be helpful to know more about such kind of biases that might affect volunteers as a whole. It was helpful to have your excellent statistical analysis. The timing especially with comparing non-Covid times to Covid times especially for a volunteer psychology projects seems like it would have added additional confounding factors to the outcome as you have pointed out. It would be interesting to repeat such a study at the same time and with fewer variables being measured. 

Thank you for your feedback. We totally agree that repeating all studies in the same time would be more that highly advisable.

---

## [Editor Report · Decision Letter 1]

17 Jan 2023

Self-selection biases in psychological studies Personality and affective disorders are prevalent among participants

PONE-D-22-27998R1

Dear Dr. Kaźmierczak,

We’re pleased to inform you that your manuscript has been judged scientifically suitable for publication and will be formally accepted for publication once it meets all outstanding technical requirements.

Kind regards,

Ankit Jain, M.D.

Academic Editor

PLOS ONE
---

## [Editor Report · Acceptance letter]

30 Jan 2023

PONE-D-22-27998R1 

Self-selection biases in psychological studies: Personality and affective disorders are prevalent among participants 

Dear Dr. Kaźmierczak:

I'm pleased to inform you that your manuscript has been deemed suitable for publication in PLOS ONE. Congratulations! Your manuscript is now with our production department. 

Kind regards, 

on behalf of

Dr. Ankit Jain 

Academic Editor

PLOS ONE